# An Emerin LEM-Domain Mutation Impairs Cell Response to Mechanical Stress

**DOI:** 10.3390/cells8060570

**Published:** 2019-06-10

**Authors:** Nada Essawy, Camille Samson, Ambre Petitalot, Sophie Moog, Anne Bigot, Isaline Herrada, Agathe Marcelot, Ana-Andreea Arteni, Catherine Coirault, Sophie Zinn-Justin

**Affiliations:** 1Sorbonne Université, INSERM UMR_S974, Center for Research in Myology, 75013 Paris, France; nada.elahmady@gmail.com (N.E.); sophie.moog@inovarion.com (S.M.); anne.bigot@upmc.fr (A.B.); 2Laboratory of Structural Biology and Radiobiology, Institute for Integrative Biology of the Cell (CEA, CNRS, University Paris South), University Paris-Saclay, 91190 Gif-sur-Yvette, France; camille.samson@pasteur.fr (C.S.); ambre.petitalot@curie.fr (A.P.); isaline.herrada@yahoo.fr (I.H.); agathe.marcelot@cea.fr (A.M.); Ana-andreea.ARTENI@i2bc.paris-saclay.fr (A.-A.A.); 3Inovarion, 251 Rue St Jacques, 75005 Paris, France

**Keywords:** emerin, atrial cardiac defects, BAF, actin, mechano-transduction

## Abstract

Emerin is a nuclear envelope protein that contributes to genome organization and cell mechanics. Through its N-terminal LAP2-emerin-MAN1 (LEM)-domain, emerin interacts with the DNA-binding protein barrier-to-autointegration (BAF). Emerin also binds to members of the linker of the nucleoskeleton and cytoskeleton (LINC) complex. Mutations in the gene encoding emerin are responsible for the majority of cases of X-linked Emery-Dreifuss muscular dystrophy (X-EDMD). Most of these mutations lead to an absence of emerin. A few missense and short deletion mutations in the disordered region of emerin are also associated with X-EDMD. More recently, missense and short deletion mutations P22L, ∆K37 and T43I were discovered in emerin LEM-domain, associated with isolated atrial cardiac defects (ACD). Here we reveal which defects, at both the molecular and cellular levels, are elicited by these LEM-domain mutations. Whereas ΔK37 mutation impaired the correct folding of the LEM-domain, P22L and T43I had no impact on the 3D structure of emerin. Surprisingly, all three mutants bound to BAF, albeit with a weaker affinity in the case of ΔK37. In human myofibroblasts derived from a patient’s fibroblasts, emerin ∆K37 was correctly localized at the inner nuclear membrane, but was present at a significantly lower level, indicating that this mutant is abnormally degraded. Moreover, SUN2 was reduced, and these cells were defective in producing actin stress fibers when grown on a stiff substrate and after cyclic stretches. Altogether, our data suggest that the main effect of mutation ΔK37 is to perturb emerin function within the LINC complex in response to mechanical stress.

## 1. Introduction

In metazoan cells, nuclei are surrounded by a nuclear envelope, containing an inner membrane and an outer membrane continuous with the membrane of the endoplasmic reticulum (ER). This nuclear envelope is rich in specific proteins, which are not normally enriched in the ER. Several of these proteins contain a LAP2-emerin-MAN1 (LEM) domain. They share the ability to bind lamins and tether repressive chromatin at the nuclear periphery. One of the best-studied LEM-domain proteins is emerin, a ubiquitously-expressed integral protein anchored at the nuclear membrane [1,2]. Emerin was first detected at the inner nuclear membrane, but is also present at the outer nuclear membrane. It consists of 254 amino acids including an N-terminal LEM-domain, a large intrinsically disordered region (IDR) and a short transmembrane segment [3,4,5]. Owing to its numerous binding partners, gathering transcription regulators as well as components of the linker of the nucleoskeleton and cytoskeleton (LINC) complex, emerin has a myriad of functions in cells. It is involved in genome organization and regulation of gene expression, as well as in signaling of mechanical stress [6,7]. Mutations in the gene coding for emerin are associated with X-linked Emery-Dreifuss muscular dystrophy (X-EDMD) [8]. These mutations are mostly nonsense, and result in the complete loss of emerin, provoking early contractures, slowly progressive humero-peroneal muscle weakness and cardiac conduction defects [9].

The absence of emerin has been implicated in a number of pathophysiological defects. For instance, in EDMD fibroblasts, non-farnesylated prelamin A is mislocalized [10,11]. During muscle differentiation, when emerin expression increases and lamin A precursor accumulates, emerin might favor the movement of non-farnesylated prelamin A into the nucleoplasm. Additionally, in mouse emerin-null myogenic progenitors, components of signaling pathways essential for myogenic differentiation and muscle regeneration are misexpressed at both the mRNA and protein levels [12,13,14]. In the hearts of emerin knockout mice, enhanced activation of the mitogen-activated protein kinase pathway is observed [15]. Increased Wnt/β-catenin signaling is associated with cardiomyocyte increased proliferation, abrogated timely cardiac differentiation as well as cardiac dysfunction [16,17]. In emerin-null fibroblasts, the centrosome is detached from the nucleus, potentially due to a loss of the interaction between emerin and tubulin [18]. The isolated nuclei of emerin-null cells show a defective nuclear adaptation to stress [19]. In the same line, emerin and non-muscle myosin IIA (NMIIA) are enriched at the outer nuclear membrane upon the application of a strain on epidermal stem cells [20]. This enrichment promotes the polymerization of perinuclear F-actin in response to stress [20]. RNAi-mediated depletion of emerin prevents strain-dependent actin polymerization around the nucleus [20].

A few emerin missense and small deletion mutations (S54F, ∆95–99, Q133H and P183H/T) have been reported in X-EDMD patients. They have been used as tools to decipher the molecular details associated with emerin functions. As they are all located in the emerin IDR (aa 50 to aa 221), this region was, thereby, described as essential for mediating emerin interaction with partners. In particular, it contributes to lamin A/C, tubulin and actin binding [18,21,22]. In addition, the IDR’s tyrosine residues are phosphorylated during nuclear adaptation to mechanical stress [19] and in cells exposed to softer extracellular matrices [23]. Such phosphorylation events contribute to mechanical stress signaling. More recently, three missense and small deletion mutations, P22L, ΔK37 and T43I, have been reported, which are associated with exclusive atrial cardiac defects (ACD) [24,25,26], Ben Yaou and Bonne, personal communication. Remarkably, all three mutations are located in the emerin LEM-domain. ΔK37 is associated with the vast majority of ACD cases (23 patients against 2 for P22L and 1 for T43I) [Ben Yaou and Bonne, personal communication]. At the molecular level and in vitro, this mutation triggers LEM-domain unfolding and favors emerin self-assembly [27]. The emerin LEM-domain binds to the highly conserved DNA-binding protein barrier-to-autointegration factor (BAF, also known as Banf1). During telophase, BAF recruits emerin to the core region of chromosomes. This recruitment is fundamental for emerin localization during nuclear assembly [15]. The crystal structures of BAF bound to either DNA or the emerin LEM-domain suggest that a dimer of BAF may simultaneously interact with two double-stranded DNA molecules and one LEM-domain [28,29]. Moreover, BAF competes with other transcription factors for emerin binding; indicating a role for emerin in gene regulation [30,31]. Finally, BAF is able to mediate interaction between emerin and the lamin A/C tail [32]. Lack of lamin A/C causes emerin to mislocalize to the cytoplasm [33]. Transient expression of the LEM-domain results in the relocation of endogenous emerin to the cytoplasm [34], suggesting that BAF is essential for emerin anchoring at the nuclear envelope through its interaction with lamin A/C.

We here present a molecular and cellular study of the impact of the three recently reported LEM-domain mutations on emerin structure and function (Figure 1a). Based on the intriguing correlation between the patient physiopathology and the site of the mutations, we hypothesized that the LEM-domain mutations possess specific functional consequences. We sought out structural defects common to the three mutants, as well as emerin functional defects in myofibroblasts derived from the fibroblasts of a patient expressing emerin ΔK37.

## 2. Materials and Methods

### 2.1. Protein Expression and Purification

Genes coding for the N-terminal octa-histidine tagged human emerin fragment EmN (region 1–187) and for human BAF with all cysteines mutated into alanine were cloned into a pETM-13 expression vector by GenScript and expressed in *Escherichia coli* BL21(DE3) cells, as formerly reported [32]. Expression vectors coding for emerin mutants ΔK37, P22L and T43I were obtained by site-directed mutagenesis (Quikchange kit, Agilent, France) from the EmN expression vector. Cultures were grown in LB broth medium for all experiments, only cultures for NMR experiments were grown in M9 minimal medium using ^15^NH_4_Cl as the sole source of nitrogen (M9 salts solution of 6 g Na_2_HPO_4_, 3g KH_2_PO_4_, 0.5 g NaCl), trace elements (26.8 µM EDTA, 6.2 µM FeCl_3_-6H_2_O, 1.24 µM ZnCl_2_, 0.152 μM CuCl_2_-2H_2_O, 0.084 μM CoCl_2_-2H_2_O, 0.324 μM H_3_BO_3_, 16.2 nM MnCl_2_-4H_2_O), 1 mM thiamine, 1 mM biotin, 300 mM CaCl_2_, 1 M MgSO_4_, 0.05% ^15^NH_4_Cl, 0.2% glucose). Cells were grown at 37 °C to an optical density (OD) of 0.8 at 600 nm and then induced with 0.5 mM isopropyl β-d-1-thiogalactopyranoside (IPTG) overnight at 20 °C. Cell pellets were suspended in 20 mL lysis buffer (50 mM Tris-HCl pH 8, 300 mM NaCl, 5% glycerol, 1% Triton TX-100 and 10 mM PMSF) per liter of culture and lysed by sonication (70% power, 4 min; pulse, 1 s; temperature, 20 °C). BAF, EmN and its mutants form inclusion bodies that were recovered from cell pellets by solubilization in 50 mM Tris-HCl pH 8, 150 mM NaCl, 20 mM imidazole, 8 M urea for at least one hour at room temperature, followed by centrifugation to remove cellular components and membranes. Supernatants were purified by affinity chromatography using NiNTA beads. The eluted proteins were refolded by dialysis overnight and two times one hour the next day (EmN and its mutants: 50 mM Tris-HCl pH 8, 30 mM NaCl; BAF: 50 mM Tris-HCl pH 8, 150 mM NaCl). Purified proteins were separated from their tags by adding the His-tagged Tobacco Etch Virus (TEV) protease. After 3h at room temperature, they were incubated with NiNTA beads, and the flow-through was dialyzed against the selected buffer.

### 2.2. Nuclear Magnetic Resonance (NMR) Spectroscopy

NMR samples containing the ^15^N-labelled proteins at 100 μM were prepared in a buffer containing 20 mM sodium phosphate pH 6.5, 30 mM NaCl, 5 mM DTT and 5% D_2_O. Two-dimensional ^1^H-^15^N HSQC experiments were recorded at 30 °C on a Bruker 750 MHz spectrometer (FMP Berlin, Berlin, Germany). All NMR data were processed using Topspin 3.1 (Bruker, Billerica, MA, USA).

### 2.3. Self-Assembly Kinetics Followed by Thioflavin T (ThT) Fluorescence

Purified EmN and its mutants were dialyzed against 20 mM Tris HCl pH 8, 30 mM NaCl, 5 mM DTT, concentrated up to 300 μM and heated at 37 °C. Their oligomerization was monitored by measuring changes of fluorescence intensity of ThT at 20 °C during 24 h. Fluorescence intensity of aliquots of protein solutions (20 μM protein and 2.5 μM ThT in 20 mM Tris HCl pH 8, 30 mM NaCl, 5 mM DTT) in 60 μL cuvette was measured at 480 nm after excitation at 440 nm using a JASCO fluorimeter equipped with an ADP-303T Peltier temperature controller (JASCO Inc., Easton, MD, USA).

### 2.4. Negative-Staining Electron Microscopy

To obtain the self-assembled state of EmN and its mutants, purified proteins were first dialyzed against 20 mM Tris-HCl pH8, 30 mM NaCl, 5 mM DTT using dry Spectra/Por dialysis membranes (6–8 kDa), then concentrated up to 500 μM, heated for 1 h at 65 °C and incubated for one week at 20 °C. Sample suspensions were applied to glow-discharged carbon-coated grids, stained with 2% *w*/*v* aqueous uranyl acetate, visualized at 100 kV with a Tecnai Spirit transmission electron microscope (FEI company, New York, NY, USA), captured by K2 4k × 4k camera (Gatan, Pleasanton, CA, USA) at 4400 or 15,000 magnification.

### 2.5. X-Ray Crystallography

The ternary complex (EmN T43I/BAF/Ig-fold domain of human lamin A/C from aa 411 to aa 566) was purified as described by Samson et al. [32]. It was concentrated to 5 mg/mL and incubated for a week at 4 °C to ensure the proteolysis of T43I, leaving only the LEM-domain of T43I bound to BAF and the lamin A/C Ig-fold domain in the complex. Hanging drop vapor diffusion was set up at 4 °C with a drop (1 μL complex solution, 1 μL reservoir solution) suspended from a glass coverslip over the reservoir solution (500 μL; 18% PEG 3350, 100 mM Tris Bis pH 5.5, 0.1 M NH_4_SO_4_). Crystals were flash-cooled in liquid nitrogen, using a cryo-protection solution (30% ethylene glycol in the reservoir solution). The 3D structure of the ternary complex was determined by molecular replacement (MolRep, CCP4i2). Coordinate files of BAF dimer bound to DNA (PDB entry 2BZF), lamin A/C globular domain (PDB entry 1IFR) and emerin LEM-domain (PDB entry 2ODC) were used to construct the structural model. This model was rebuilt by PHENIX, manual correction was performed using Coot according to |Fo| − |Fc| and 2|Fo| − |Fc| maps, and further refinement was carried out by phenix.refine. All structure figures were generated using PyMOL (Schrödinger, LLC, Cambridge, UK).

### 2.6. Size-Exclusion Chromatography (SEC)

Interactions of EmN and its mutants with BAF (EmN and its mutants at 100 μM and BAF at 200 μM in 500 μL) were identified by size-exclusion chromatography on a Superdex 75 10/300 GL column (GE Healthcare, Aulnay-Sous-Bois, France) pre-equilibrated with 20 mM Tris-HCl pH 8, 30 mM NaCl, 2 mM DTT and run at a flow rate of 0.5 mL/min at 4 °C.

### 2.7. Cell Culture and Reagents

Stable MyoD transductions were performed in human immortalized fibroblasts using a doxycycline-inducible Myod1 lentivirus. Single wild-type and ΔK37 clones were used in this study, both of which produced MyoD after doxycycline induction in 100% of the cells. MyoD-transfected fibroblasts were cultured in a proliferation medium consisting of Dulbecco’s Modified Eagle Medium (DMEM), supplemented with 10% fetal bovine serum (Life Technologies, Villebon Sur Yvette, France) and 0.1% gentamycin (Invitrogen, Paris, France). For myoconversion, doxycycline (2 µg/mL; Sigma Aldrich, Saint-Quentin-Fallavier, France) was added in the differentiation medium, composed of DMEM with 10 µg/mL Insulin. All cells were grown in a humidified cell culture incubator at 37 °C and 5% CO_2_. All cells were grown in a humidified cell culture incubator at 37 °C and 5% CO_2_. All cells were mononucleated when analyzed.

### 2.8. Cell Plating on Substrates of Different Stiffness

A commercially-available soft substrate of 8 kPa stiffness (Matrigen Brea, CA, USA) was used. For hard substrates, cells were plated on glass slides. All substrates were coated with fibronectin at a concentration of 10 µg/mL (Sigma-Aldrich).

### 2.9. Cyclic Strain

Cells were plated on tissue train plates (Flexcell International, Asbach, Denmark) coated with GFR Matrigel (Matrigel Matrix, Corning, Life Sciences, Illkrich, France) for one day and stretched (10% elongation, 0.5 Hz, 4 h). Cells were then collected for subsequent experiments. Control cells were collected under the same experimental conditions, without stretching.

### 2.10. Antibodies

Fixed cells were stained using the antibodies listed in Table 1 and the following reagents: F-actin was stained with Alexa Fluor 568 Phalloidin (Thermofisher, Les Ullis, France) and nuclei were counter-stained with Hoechst 33,342 (H3570, Thermofisher). Secondary antibodies for immunoblotting were HRP-conjugated goat anti rabbit, rabbit anti-mouse, donkey anti-goat IgGs (Jackson ImmunoResearch, Cambridge, UK). Secondary antibodies for immunofluorescence were Alexa Fluor-488 or 568 conjugated goat anti-rabbit IgG, Alexa-Fluor-568-conjugated donkey antigoat IgG, Alexa Fluor 488 or 568-conjugated goat anti-mouse IgG (Life Technologies).

### 2.11. Immunohistochemistry and Immunofluorescence Microscopy

Cells were first fixed (4% paraformaldehyde in phosphate-buffer saline (PBS), for 5 min at room temperature (RT)), then permeabilized (with 0.1% Triton X-100 in PBS, for 5 min, at RT), before blocked (with 10% BSA in PBS, for 2 h, at 4 °C). Cells were incubated with primary antibodies (in PBS with 5% BSA, overnight) and washed next in PBS. All secondary antibodies were first incubated for 1 h at RT in PBS, then washed, after that nuclei were counterstained with Hoechst. Slides were mounted on Mowiol mounting medium. Immunofluorescence microscopy was carried using an Olympus FV1000 (Olympus, Rungis, France) or a laser-scanning microscopy Nikon Ti2 coupled to a Yokogawa CSU-W1 head for confocal images.

### 2.12. Protein Extraction and Immunoblotting

Total proteins were extracted in cell lysis buffer (2% SDS, 250 mM sucrose, 75 mM urea, 50 mM Tris HCl pH 7.5, 1 mM DTT) with the addition of Complete^®^ protease inhibitors (Roche, Boulogne-Billancourt, France). Lysates were sonicated (three pulses of 10 s at 30% amplitude), spin extracted (at 14,000 rcf, 10 min, 4 °C) and quantified using a bicinchoninic acid assay (BCA; Thermo Fischer Scientific). Extracts were separated by 10%–12% SDS-PAGE and transferred onto 0.45 μm nitrocellulose membranes (Invitrogen). Membranes were blocked in 5% low-fat milk in TBS-Tween20 (1 h, RT), then incubated with the selected primary antibody (overnight at 4 °C or for 4 h at RT). Membranes were first washed with TBS-Tween20 before incubation with secondary horseradish peroxidase (HRP)-conjugated anti-rabbit or anti-mouse or anti-goat antibodies (for 1 h, at RT). Signals were revealed using an Immobilon Western Chemiluminescent HRP Substrate (Millipore, Guyancourt, France) on a G-Box system with GeneSnap software (version 7.02, Cambridge, UK). ImageJ software (version 1.49, National Institutes of Health, Bethesda, MD, USA) was used for quantification of band intensities.

### 2.13. qPCR

An RNase mini-kit (Qiagen, Courtabeuf, France) was used to prepare total RNA. For reverse transcription and quantitative RT-PCR, Superscript III (Life technologies) with random primers was used for cDNA generation and SYBR Green PCR Master Mix was used according to the manufacturer instructions. Experiments were performed on a Light Cycler 480 System (Roche), with each sample performed in triplicate. Ribosomal Protein Lateral Stalk Subunit P0 (*RPLP0*) was used as housekeeping gene. Primer sequences are listed in Table 2.

### 2.14. Image Analysis

All image analyses were performed using FIJI (version 1.52, National Institutes of Health, Bethesda, MD, USA). Actin fiber numbers were determined by drawing a line perpendicular to the long axis of the nucleus. Then, plot profiles were analyzed using the command “Find Peaks”, where the number of peaks represented the number of actin fibers. Nuclear envelope thickness and fluorescence intensity were measured on confocal images by drawing a 10 µm-line perpendicular to the long axis of the nucleus.

### 2.15. Statistics

GraphPad Prism software (version 5.04, San Diego, CA, USA) was used to calculate and plot mean and standard error of the mean (SEM) of measured quantities. Statistical significance was assessed by an ANOVA followed by a Bonferroni or two-tailed unpaired Student’s *t*-test. *p* values < 0.05 were considered as significant.

## 3. Results

### 3.1. The Three Emerin Mutations Associated With Cardiac Defects Favor Emerin Self-Assembly In Vitro

We first analyzed the impact of the three LEM-domain mutations on the structural properties of the emerin nucleoplasmic region. Since these mutations are all located in the only globular domain of emerin (aa 1 to aa 45; Figure 1a), we hypothesized that they could destabilize the LEM-domain. We had previously shown that the mutation ΔK37 indeed causes a loss of the 3D structure of this domain, as observed using NMR by analyzing the 2D ^1^H-^15^N HSQC spectrum of the construct EmN (aa 1 to aa 187) containing the mutation ΔK37 [27]. Here, we produced the ^15^N-labeled EmN mutants P22L and T43I, and we recorded their 2D ^1^H-^15^N HSQC spectra. These spectra nicely overlap with the 2D ^1^H-^15^N HSQC spectrum of wild-type EmN (Figure 1b). Only a few signals corresponding to the residues close to the site of mutation are shifted, demonstrating that the mutations P22L and T43I do not modify the 3D structure of the emerin construct EmN. The LEM-domain is involved in emerin self-assembly [4]. Its interaction with the EmN disordered region triggers the formation of curvilinear filaments in vitro [27,35]. We, thereby, tested if the three LEM-domain mutations impair the self-assembly of EmN. We showed, using a thioflavin T fluorescence assay, that the three mutated EmN polymerize significantly faster than wild-type EmN in vitro (Figure 1c). Furthermore, negative-staining electron microscopy images of the self-assembled mutated EmN revealed filaments undistinguishable from those obtained with the wild-type EmN (Figure 1c; [27]). Thus, although only mutation ΔK37 significantly modifies the 3D structure of the LEM-domain, all three mutations favor EmN self-assembly in vitro.

### 3.2. Mutation ΔK37, Most Commonly Found in Patients With ACD, Causes Emerin Degradation in the Cell

Amongst the three emerin mutations studied here, ΔK37 was the most frequently reported in patients with ACD (23 patients against 2 for P22L and 1 for T43I; [24,25,26], Ben Yaou and Bonne, personal communication. We further analyzed the impact of this mutation on emerin expression and function in myofibroblasts, which were derived from a patient’s fibroblasts after the overexpression of MyoD. Figure 2a shows that emerin ΔK37 was correctly localized at the nuclear membrane, as described for wild-type emerin. Quantification of emerin staining thickness at the nuclear membrane showed no change due to the mutation (Figure 2b). However, the staining intensity at the nuclear membrane was significantly weaker in the case of ΔK37 (Figure 2a), as quantified in Figure 2c. We measured emerin mRNA levels in cells expressing either wild-type or mutated emerin. There was no significant difference between the two cell lines (Figure 2d). However, emerin protein quantification by Western blot confirmed that emerin was less abundant in cells expressing emerin ΔK37 compared to the wild-type (Figure 2e). Altogether our data suggest that emerin ΔK37 degraded faster than the wild-type emerin in myofibroblasts.

### 3.3. The Three Emerin Mutants Interact With BAF in Vitro

The best-characterized LEM-domain function is to mediate emerin interaction with chromatin [6,36]. Indeed, this emerin LEM-domain directly interacts with the barrier-to-autointegration factor (BAF), an 89-residue protein that binds to double-stranded DNA [37]. Analysis of the 3D structure of the LEM-BAF complex [28] reveals that emerin Pro22 and Lys37 belonged to a well-conserved surface that is involved in BAF binding (in blue on Figure 3a). Both residues contributed to the interface with BAF. By contrast, Thr43 mutation was located on the opposite side of the LEM-domain, far from the interface with BAF (Figure 3a). Therefore, we hypothesized that mutations P22L and ΔK37 interfere with BAF binding, whereas mutation T43I has no impact on BAF binding. In order to test our hypotheses, we produced the three mutated EmN fragments, as well as the BAF dimer. We then examined, using size-exclusion chromatography (SEC), the interaction of the mutants with BAF. Unexpectedly, we observed that all three mutants bound to BAF (Figure 3b). With that said, in the presence of EmN ΔK37, the elution of BAF was delayed compared to what was observed in the presence of wild-type EmN and the two other mutants, suggesting that EmN ΔK37 had a weaker affinity for BAF. Finally, we tried to crystallize the ternary complexes formed by the EmN mutants, BAF and the Igfold domain of lamin A/C using the same protocol used for wild-type EmN [32]. We succeeded in solving the 3D structure of the complex containing EmN T43I at a resolution of 2.3 Å (Appendix A). The structure of the mutated LEM-domain bound to the BAF dimer is displayed on Figure 3c. It was highly similar to the structure of the wild-type complex: The root-mean-square deviation for the Cα atoms was 0.55 Å. This signifies that mutation T43I did not influence emerin binding to BAF.

### 3.4. Mutation ΔK37 Does Not Impact Levels of Lamin A/C and SUN1 Nor Their Localization, But Causes a Significant Decrease in SUN2 Level

The reduced expression of emerin observed in mutant ΔK37 cells may impact the localization of its binding partners, namely BAF, lamin A/C [32], SUN1 and SUN2 [38]. To assess this hypothesis, we first checked BAF protein localization in ΔK37 myofibroblasts. Figure 4a shows that BAF was mainly localized in the nucleus of cells, with no obvious difference between wild-type and ΔK37. Likewise, lamin A/C was distributed at the nuclear membrane in both cell lines (Figure 4b). Quantification of the amount of lamin A/C by Western blot did not indicate any significant differences between these cell lines (Figure 4c,d). SUN2 was localized at the nuclear rim in both wild-type and ΔK37 (Figure 4e). Conversely, SUN2 staining appeared weaker in ΔK37 (Figure 4e). Quantification of the amount of SUN2 by Western blot revealed that SUN2 was significantly lower in ΔK37 compared with wild-type cells (Figure 4c,d). In contrast, no obvious difference in SUN1 immunostaining (Figure 4f) or Western blot quantification (Figure 4c,d) was observed between these cell lines.

### 3.5. Mutation ΔK37 Impairs the Cell Response to Substrate Stiffness and Cyclic Stretch

Recent studies have highlighted the contribution of emerin, together with lamin A/C, SUN1 and SUN2, to the signaling of mechanical stress between the cytoskeleton and the nucleus [19,20,39]. As we observed that emerin and SUN2 levels are lowered in cells expressing emerin ΔK37, we hypothesized that this could affect how the cells sense and respond to mechanical cues. First, we examined the organization and the number of actin fibers present at the bottom and the top of the nuclei in our myofibroblasts (Figure 5a). When the cells were grown on a rigid substrate, i.e., glass, the number of actin fibers counted at the top of the nuclei was lower and less organized in the cells expressing emerin ΔK37 than in wild-type cells (Figure 5a,b). In contrast, no significant difference was detected in the number of supranuclear actin fibers when the two cell lines were grown on a soft substrate (Figure 5b). Our quantitative analysis demonstrated that both cell lines have similar spreading areas and they increase these spreading areas when grown on a hard substrate as opposed to a soft substrate (Figure 5c). To further explore whether cells expressing the emerin mutant are defective in triggering the assembly of actin stress fibers in response to mechanical stress, cells were exposed to cyclic stretches (Figure 6). Quantification of the number of actin stress fibers induced by the cyclic stretching events established that, while cells expressing wild-type emerin respond to cyclic stretches by increasing their number of stress fibers, this increase is significantly lower in cells expressing emerin ΔK37 (Figure 6b). In contrast, the cyclic stretches did not affect the nuclear shape in either cell lines (Figure 6c). In the same experimental setup, we also analyzed the cytoplasmic localization of emerin using digitonin permeabilization. In wild-type cells, emerin is mainly nuclear before stretch and is partially exported into the cytoplasm after stretch, as reported elsewhere [20] (Figure 6a). Myofibroblasts expressing ΔK37 did not show obvious defects in nuclear export, emerin being observed in the cytoplasm both, at baseline and after cyclic stretches. Nevertheless, due to the weak emerin staining in these cells, it is difficult to conclude regarding a potential stretch-induced nuclear export of emerin ΔK37. Finally, we investigated the capacity of the mutated emerin to be phosphorylated by Src. Indeed, it was demonstrated that in response to mechanical force application on isolated nuclei, emerin tyrosines 74 and 95 are phosphorylated by Src, as part of the mechanical stress signaling pathway [19]. SDS-PAGE analysis of His-tagged EmN ΔK37 phosphorylated by Src in vitro, suggests that it is phosphorylated to a similar level as wild-type His-tagged EmN (Figure 6d). We conclude that ΔK37 does not significantly impair the phosphorylation of EmN by Src. Our results, taken together, revealed that whereas cells expressing emerin ΔK37 showed significant defects in cytoplasmic actin plasticity, the other parameters studied—nuclear mechanics, emerin phosphorylation by Src and cytoplasmic emerin localization after mechanical stress—were preserved in these cells.

## 4. Discussion

Emerin is mutated in a large number of patients with EDMD including cardiac defects [3,8,40], and in a smaller number of patients with exclusive atrial cardiac defects [28,31,41], Ben Yaou & Bonne, personal communication. Missense mutations (as S54F, D72V, Q133H, P183H/T) and a small deletion (Δ95–99) associated with EDMD were identified in the disordered region of emerin, required for its interaction with lamin A/C, tubulin and actin [42,43,44,45]. Interestingly, missense mutations (P22L, T43I) and a small deletion (ΔK37) associated with exclusive cardiac defects were detected in emerin LEM-domain that interacts with the DNA binding protein BAF [28,31]. This striking correlation between the patient physiopathology and the protein structure raised the possibility that the LEM-domain mutations hold cardioselective implications. For that reason, we were prompted to (i) identify in vitro molecular defects common to the three LEM-domain mutations, and (ii) search for cellular defects in myofibroblasts derived from a patient’s fibroblasts expressing emerin ΔK37. As the only known protein partner of emerin LEM-domain is the DNA-binding protein BAF, we expected that the LEM-domain mutations associated with cardiac defects would impair BAF binding. This hypothesis was supported by a previous study, in which we demonstrated that the LEM-domain of emerin ΔK37 loses its α-helical 3D structure, as observed by NMR [27]. It was also supported by a study from K. Wilson and co-workers who showed that targeted mutagenesis of the LEM-domain results in reduced emerin-BAF binding in a biochemical assay [21]. Yet, surprisingly, we evidenced that (i) the two other LEM-domain mutants, P22L and T43I, have a 3D structure similar to that of the wild-type emerin, (ii) all three LEM-domain mutants bind to BAF. The mutant ΔK37 is able to transiently form a BAF-binding competent structure, which interacts with the BAF dimer through an interface that we fell short to capture using X-ray crystallography. It exhibits a small but significant reduction in affinity for BAF compared to wild-type emerin, suggesting that indeed, destabilization of its 3D structure impacts BAF binding. Analysis of the crystal structure of EmN T43I bound to BAF corroborated that this mutation does not perturb the interface with BAF.

Emerin oligomerizes at the inner nuclear membrane [4,35]. In vitro, emerin self-assembly regulates its binding properties: only self-assembled emerin is able to directly interact with the lamin A/C tail [32]. We revealed that, in vitro again, all three mutated EmN fragments polymerize significantly faster than wild-type EmN. This suggests that the self-assembly of the LEM-domain mutants is altered in cells. Additionally, we observed a reduced emerin ΔK37 protein expression without any modification of emerin mRNA levels, indicating that emerin ΔK37 is either less transcribed because of defective post-transcriptional regulation of emerin mRNA, or degraded faster than wild-type emerin in myofibroblasts. A lower level of emerin ΔK37 has been consistently reported in lymphoblastoid cell lines and skin fibroblasts from female and male patients [25] and a near-total lack of nuclear staining was described in buccal cells from male patients [26]. In the same line, a lack of emerin P22L staining has been reported in a male patient deltoid muscles [24]. From all these data, we propose that the LEM-domain mutations impact emerin self-assembly and cause emerin degradation in cells.

Emerin ΔK37 is correctly localized in myofibroblasts derived from patient fibroblasts. It was reported that emerin localization at the inner nuclear envelope depends on the presence of lamin A/C at the nuclear periphery [33,46]. In particular, in the skin fibroblasts obtained from a male child homozygous for the Y259X LMNA mutation, no lamin A/C is detected and emerin exhibits aberrant localization in the endoplasmic reticulum [33,46,47]. Moreover, nuclei are morphologically abnormal. In this report, we highlighted that lamin A/C is present at the nuclear envelope of cells expressing emerin ΔK37. Moreover, lamin A/C is able to bind to emerin ΔK37 through BAF. These two observations are consistent with the wild-type localization of emerin ΔK37. The lamin wild-type level and localization also might explain why no nuclear deformation was detected, even after the cyclic stretching of the cells.

We showed that the mutants are still able to interact with BAF, and thus can still be associated with chromatin, regulating gene expression in muscle tissues. An extensive analysis of gene expression in differentiating emerin-null myogenic progenitors has pointed out that emerin functions during the transcriptional reprograming of progenitors to committed myoblasts [12]. That being the case, the availability of myofibroblasts expressing each of the emerin LEM-domain mutants would facilitate a transcriptomic analysis, focused on the molecular pathways implicated in muscle cell differentiation. This could eventually substantiate that even a low level of emerin is sufficient to fulfill its essential functions in gene expression regulation during myoblast differentiation.

We revealed that the less abundant emerin ΔK37 causes defects in mechanical stress signaling. Lammerding et al. previously reported that emerin-deficient mouse embryo fibroblasts have an apparent normal mechanics, but an impaired expression of mechanosensitive genes in response to strain [48]. Le et al. revealed that the force-dependent emerin enrichment at the outer nuclear membrane triggers the recruitment of non-muscle myosin IIA and the formation of a perinuclear F-actin ring [20]. In human keratinocytes treated with silencing RNA targeting emerin, they observed a reduction of strain-induced F-actin accumulation around the nucleus [20]. Although patients bearing our LEM-domain mutations exhibit exclusive atrial cardiac defects, we consistently observed a defective formation of actin stress fibers when myofibroblasts expressing emerin ΔK37 were grown on a hard substrate or subjected to cyclic stretches. Thus, a decrease in the emerin protein level might be the common cause for the lack of strain-induced actin stress fibers in cells of patients with atrial cardiac defects due to mutations in the *EMD* gene coding for emerin. On the other hand, our results showed that the protein level of SUN2, a member of the SUN domain protein family that typically perform their functions within nuclear envelope-spanning LINC complexes, was reduced in ΔK37. Decreased SUN2 expression could result from inhibition of gene expression or targeted protein degradation. Further studies are required to determine detailed protein regulation of SUN2 in these cells. The LINC complexes connect the perinuclear actin fibers to the nuclear envelope [49] and tension in actin stress fibers leads to a local enrichment of LINC complexes and lamin A/C [50]. Moreover, it has been recently suggested that SUN2 signaling, from the nuclear envelope to the cytoplasm, favors RhoA activation, promoting assembly of actin stress fibers [51]. Hence, SUN2 depletion could contribute to the failure to properly assemble perinuclear actin fibers, which are critical for heart muscle function. Alternatively, SUN2 plays prominent roles in the resistance to DNA damage [52]) and acts as anti-fibrogenesis factor, at least in the liver [53]. It will now be interesting to determine whether SUN2 deficiency promotes cardiac fibrosis, as well as to explore the expression of SUN2 in other experimental models of emerinopathies.

## 5. Conclusions

Using diverse biochemical and cellular approaches, we investigated three emerin LEM-domain mutations present in patients with an isolated atrial cardiac disease. A previous study showed that the mutation ΔK37 results in a loss of emerin 3D structure. Here we made the unexpected observation that two other mutations, P22L and T43I, did not modify the LEM-domain structure, and that none of the three mutations hinder LEM-domain binding to its nuclear partner BAF. This suggests that the nuclear functions of emerin could be fulfilled by the LEM-domain mutants. Furthermore, lamin A/C was present in myofibroblasts expressing emerin ΔK37. No nuclear shape defect was observed, both on soft and hard substrates, as well as before and after cyclic stretches. However, emerin mutants showed an excessive propensity to self-assemble in vitro and are degraded in patient cells, and in our derived myofibroblasts. The level of SUN2 is also significantly decreased in ΔK37. The low abundance of both emerin and SUN2 may impair the formation of F-actin stress fibers, when myofibroblasts expressing emerin ΔK37 are grown on glass or exposed to cyclic stretches. Subtle defects in cytoplasmic remodeling may particularly affect tissues that routinely experience rhythmic mechanical strains throughout life, such as the cardiac muscle.

## Figures and Tables

**Figure 1 cells-08-00570-f001:**
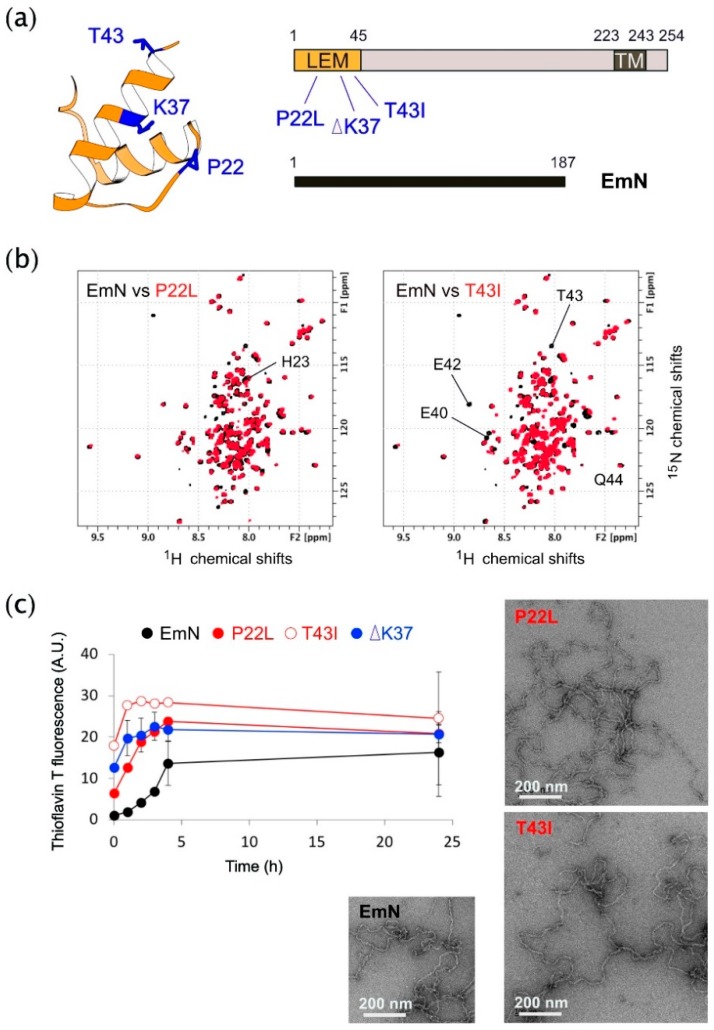
The three emerin mutations associated with cardiac defects favor emerin self-assembly in vitro. (**a**) Emerin structure. Left panel: Ribbon diagram of emerin LAP2-emerin-MAN1 (LEM)-domain in orange with proline 22, lysine 37 and threonine 43 in blue. Right panel: Representation of full length emerin (residues 1–254), and the fragment EmN (residues 1–187). Full length emerin is composed of 254 residues. It exhibits an N-terminal 45-residue LEM-domain, including P22, K37 and T43, a transmembrane domain (residues 223–243) and a C-terminal luminal domain (residues 244–254). (**b**) Impact of LEM domain mutations on emerin conformation in solution. Superimposition of the 2D NMR ^1^H-^15^N HSQC spectra of wild-type EmN (in black) with mutants P22L (left panel) or T43I (right panel), both in red. All spectra were recorded at 30 °C, 750 MHz and 150 µM. The spectra of the two mutants nicely overlay with that of EmN wild-type. The few shifted peaks, corresponding to H23 in P22L, and E40, E42, T43 and Q44 in T43I, are indicated. (**c**) EmN and variants self-assembly kinetics. Left panel: Thioflavin T (ThT) fluorescence was measured as a function of the incubation time at 37 °C for EmN wild-type, P22L, ΔK37 and T43I (black, filled red, blue and open red symbols, respectively). All proteins were at a concentration of 300 µM during the kinetics and 20 µM during the fluorescence measurements. Right panel: Self-assembled P22L and T43I (500 µM) were heated for 1 h at 65 °C, then incubated for one week at 20 °C. Analysis by negative stain imaging demonstrates the formation of curvilinear filaments of 10 nm diameter. A negative-staining image of filaments obtained from EmN is displayed for comparison. Scale bar, 200 nm.

**Figure 2 cells-08-00570-f002:**
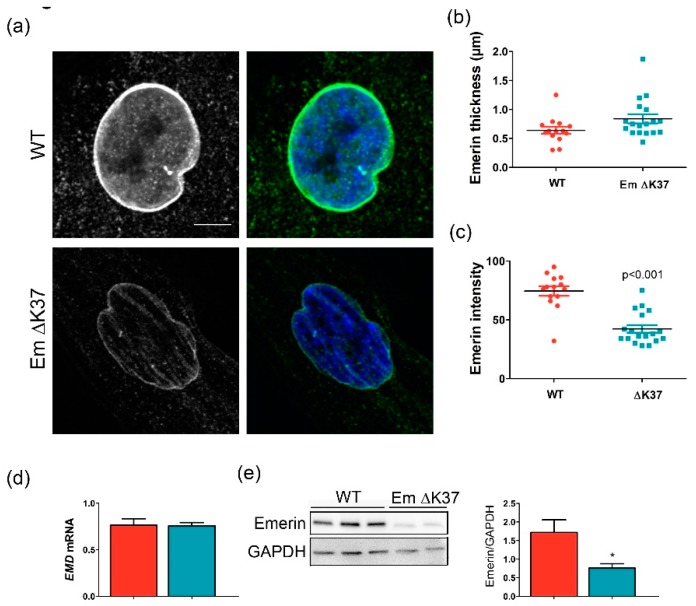
Mutation ΔK37, most commonly found in patients with atrial cardiac defects (ACD), causes emerin degradation in cells grown on hard substrate. (**a**) Representative immunofluorescence images of emerin localization in wild-type (WT) and ΔK37 myofibroblasts. DNA is stained in blue using Hoechst 33,342 and emerin in green. Scale bar, 10 µm. (**b**) Scatter plots of emerin nuclear envelope staining thickness (μm) in wild-type (red) and ΔK37 (blue) myofibroblasts. (**c**) Scatter plots of emerin nuclear envelope staining intensity in wild-type (red) and ΔK37 (blue) myofibroblasts. (**d**) Emerin mRNA expression in wild-type (red) and ΔK37 (blue) myofibroblasts. *EMD* gene expression was normalized to *RPLP0*. Data represent mean and SEM (*n* = 3 for each cell line). (**e**) Typical western-blot of emerin protein expression in wild-type and ΔK37 myofibroblasts, and mean emerin protein levels in wild-type (red) and ΔK37 (blue) myofibroblasts (normalized to Glyceraldehyde-3-Phosphate Dehydrogenase, GAPDH). Data represent mean and SEM (*n* = 7 in WT, *n* = 6 in ΔK37). * *p* < 0.05.

**Figure 3 cells-08-00570-f003:**
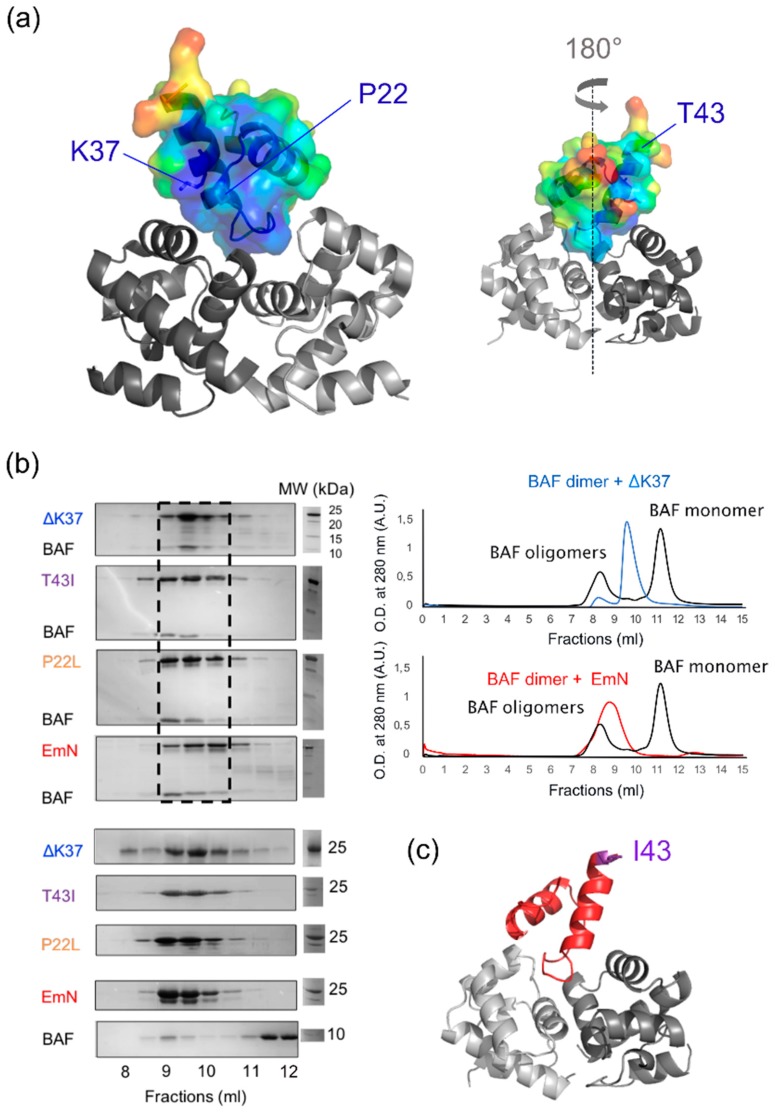
The three emerin mutants interact with the barrier-to-autointegration factor (BAF) in vitro. (**a**) Molecular surface representation of the LEM-domain (color gradient) docked to its binding site within the BAF dimer (grey cartoon), in two orientations (180° rotation, as indicated). Lysine 37, proline 22 and threonine 43 are indicated with blue arrows. Color gradient represents LEM sequence conservation (highly conserved in blue to poorly conserved in red). (**b**) Size-exclusion chromatography was performed on EmN wild-type and mutants, in the absence and presence of BAF, in order to obtain reference and binary-complex chromatograms, respectively. In all experiments, proteins were injected at the same concentration (100 µM EmN, 200 µM BAF), using the same buffer and column (Superdex 75 10/300 GL). Left panel: SDS-PAGE gels corresponding to references and binary-complex chromatograms are presented, from bottom: References for BAF (black), EmN (red), P22L (orange), T43I (purple) and ΔK37 (blue) followed by EmN-BAF, P22l-BAF, T43I-BAF and ΔK37-BAF binary complexes. Bands corresponding to the four binary complexes are boxed (black dotted rectangle). Right panel: Chromatograms representing the elution fractions of BAF alone are shown in black, binary complexes of ΔK37-BAF in blue and wild-type EmN-BAF in red. (**c**) The 3D structure of the LEM-domain (red) of mutant T43I (threonine 43 in purple) bound to the BAF dimer (grey) resolved by X- ray crystallography at a resolution of 2.3 Å (conditions: 18% PEG 3350, 100 mM Tris Bis pH 5.5, 0.1 M NH_4_SO_4_; Appendix A).

**Figure 4 cells-08-00570-f004:**
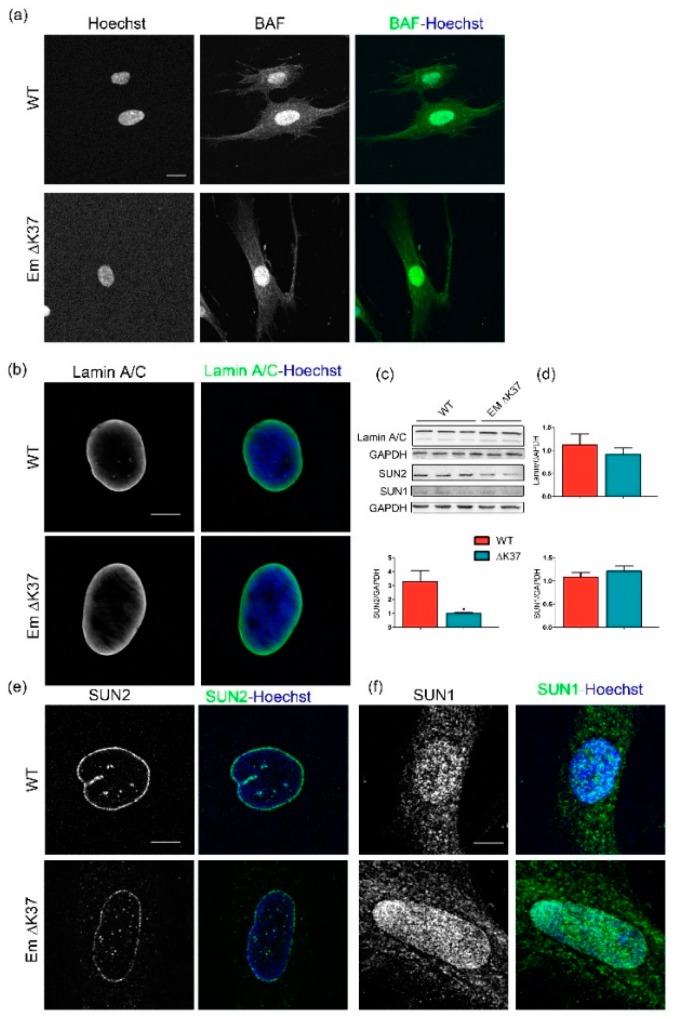
ΔK37 mutation does not impact the localization of BAF, localization and level of lamin A/C and SUN1, but causes a significant reduction in SUN2 in cells grown on hard substrate. (**a**) Representative images of wild-type and ΔK37 myofibroblasts stained for BAF (green); nuclei (blue); scale bar, 20 µm. (**b**) Representative images of wild-type and ΔK37 myofibroblasts stained for lamin A/C (green); nuclei in blue; scale bar, 30 µm. (**c**) Representative western-blot of lamin A/C, SUN1 and SUN2 in wild-type and ΔK37 myofibroblasts. GAPDH was used as an internal control. (**d**) Protein levels of lamin A/C (*n* = 10 in wild-type, *n* = 8 in ΔK37), SUN1 (*n* = 5 in wild-type, *n* = 4 in Δ K37) and Sun2 (*n* = 7 in wild-type, *n* = 5 in ΔK37) in wild-type (red) and ΔK37 (blue) myofibroblasts, as quantified from the western blot assays; * *p*-value < 0.05. (**e**) Representative images of wild-type and ΔK37 myofibroblasts stained for SUN2 (green); nuclei in blue; scale bar, 30 µm. (**f**) Representative images of wild-type and ΔK37 myofibroblasts stained for SUN1 (green); nuclei in blue; scale bar, 30 µm.

**Figure 5 cells-08-00570-f005:**
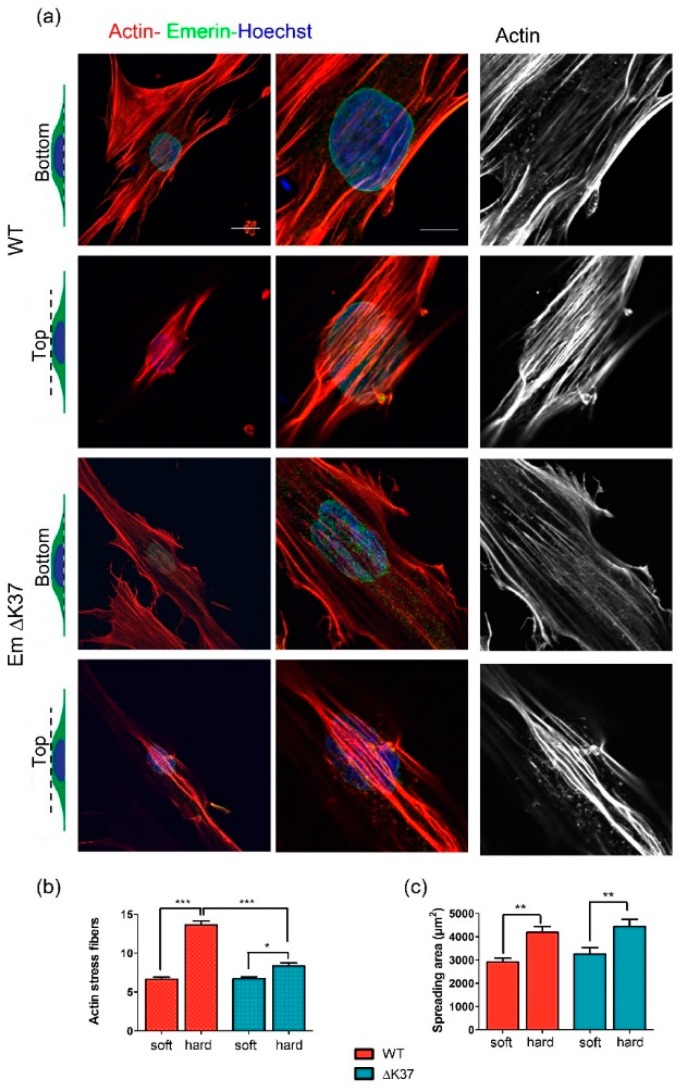
Mutation ΔK37 impairs perinuclear actin organization on cells grown on a hard substrate. (**a**) Representative images of wild-type (upper panel) and ΔK37 (lower panel) myofibroblasts, plated on 2D hard substrate stained for F-actin (phalloidin, red), emerin (green) and nuclei (Hoescht, blue). Sections of actin filament network are shown at basal and apical surfaces of the cell, as indicated on the left. Scale bar, 20 µm, or 10 µm (zoom). (**b**) Quantification of perinuclear actin stress fibers in wild-type (red) and ΔK37 (blue) myofibroblasts on soft (8 kPa) and hard (glass) substrates. Data represent mean ± SEM. Mean number from *n* = 30 ± 1 cell for each group from two independent experiments. * *p* < 0.05, *** *p* < 0.01. *n* = 0.001. (**c**) Quantification of wild-type (red) and ΔK37 (blue) myofibroblasts spreading areas on soft (8 kPa) and hard (glass) substrates. Data represent mean ± SEM. ** *p* < 0.01. *n* > 33 cells for each group from two independent experiments.

**Figure 6 cells-08-00570-f006:**
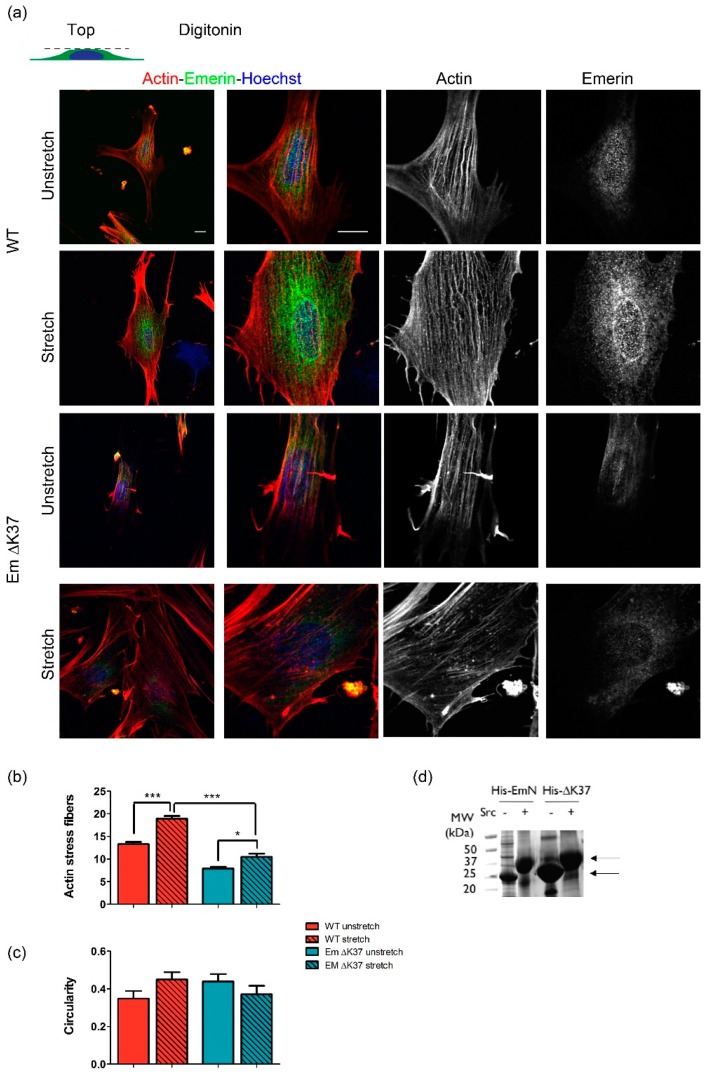
Mutation ΔK37 impairs the perinuclear actin response to cyclic stretch. (**a**) Representative images of wild-type (upper panels) and ΔK37 (lower panels) myofibroblasts stained for F-actin (phalloidin, red), emerin (green) and nuclei (Hoechst, blue) in unstretched and stretched conditions. The outer nuclear membrane was permeabilized using digitonin and cells were subjected to 10% cyclic stretch at 0.5 Hz for 4 h. Scale bar, 10 µm. (**b**) Number of actin stress fibers in wild-type (red) and ΔK37 (blue) myofibroblasts in unstretched (clear) and stretched (striated) conditions. Data represent mean ± SEM from 30 ± 1 cells for each condition in two independent experiments. * *p* < 0.05, *** *p* < 0.001 versus WT. (**c**) Contour ratio (4π area/perimeter^2^) as a measurement of nuclear circularity, calculated for wild-type (red) and ΔK37 (blue) myofibroblasts in unstretched (clear) and stretched (striated) conditions. Data represent mean ± SEM in 30 ± 1 cells for each condition in two independent experiments. (**d**) Phosphorylation of 8His-EmN and 8His-ΔK37 (0.9 mg) by Src kinase (0.9 μg) in 2 mM ATP, 0.5 mM MgCl_2_, 5 mM DTT, 20 mM Tris HCl pH 7.5, protease inhibitor, overnight at 30 °C. The SDS-PAGE gel shows 8His-EmN and 8His-ΔK37 before (−), and after phosphorylation (+). Two arrows indicate where EmN migrates before and after phosphorylation.

**Table 1 cells-08-00570-t001:** List of antibodies used in this study.

Name	Reference	Host	Supplier
Anti-emerin	ab40688	Rabbit	Abcam
Anti-lamin A + C [131C3]	ab8984	Mouse	Abcam
Anti-SUN2 [EPR6557]	ab124916	Rabbit	Abcam
Anti-BANF1/BAF [EPR7668]	ab129184	Rabbit	Abcam
Anti-SUN1	C3286	Rabbit	Generously provided as a gift from D. Hodzic, Department of Ophthalmology and Visual Sciences, Washington University School of Medicine, St Louis, MO, USA

**Table 2 cells-08-00570-t002:** List of qRT-PCR primers used in this study.

Gene	Forward (5′-3′)	Reverse (5′-3′)
*RPLPO*	CTCCAAGCAGATGCAGCAGA	ATAGCCTTGCGCATCATGGT
*EMD*	CCCTGCCAGCCAGTCCCCTCG	CACCCCCACTGCTAAGGCAGTCAGC

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
