# Peer review of "An Emerin LEM-Domain Mutation Impairs Cell Response to Mechanical Stress"

_cells, 2019, doi:10.3390/cells8060570_

Round 1

Reviewer 1 Report

The paper by Essawy et al. is invetigating the effects of three mutations within the LEM domain of emerin. This paper is overall well written and conclusive.

I found only one minor issue that needs to be changed. In figure 4a it is not labelled which is the mutant and which is wt. Also the claim that BAF is mainly localized in the nucleus can't be proven by the figure as it is, the Hoechst staining covers the nuclear area making it impossible to see/compare BAF staining there. I recommend a higher magnification and a splitting of channels (each in grey, plus a merge) for figure 4a.

Author Response

The authors greatly thank Reviewer 1 for her/his helpful comments. All comments have been taken into account.

The Figure 4a has been modified as recommended.

Reviewer 2 Report

Journal: Cells

Manuscript ID: cells-516098

The manuscript titled, “An emerin LEM-domain mutation impairs cell response to mechanical stress” submitted by Essawy et al examines the functional consequences of several mutations of the nuclear envelope protein emerin using a combination of structural chemistry, molecular and cellular biology, and biochemical techniques. Their data indicate that of the 3 emerin mutations typically associated with clinical cardiac defects, it is the DK37 mutation that results in down regulated expression of emerin which negatively impacts the formation of actin stress fibers in myofibroblasts. This is significant given the biomechanical stresses imposed on cells within the heart. The authors go on to conclude that while the emerin DK37 mutation does not impair its ability to interact with its endogenous binding partner, BAF, or affect morphology of the nuclear lamina, its increased propensity to self-assemble as well as constitutive low protein expression affects expression of LINC components that facilitate proper filamentous actin stress fiber formation.

Overall, this is a well written paper that provides novel functional insights into a nuclear envelope protein and how mutant forms of it impairs cell morphology. Furthermore, these data imply that similar signaling may exist in cardiac cells, potentially advancing knowledge of the mechanistic role of emerin in cardiac disease.

There are a few minor comments I feel that the authors could address to strengthen their manuscript prior to acceptance, detailed below:

General comments:

1)      There are a few syntax errors that interrupt the flow of the manuscript. For example, in the legend for Figure 1, it is written, “The three emerin mutations associated to cardiac defects…” should probably be revised to read, “The three emerin mutations associated with cardiac defects…”

2)      At least one paragraph in the introduction section reads more like it would fit in the Discussion section of the paper, i.e. – pg. 2, lines 49-64. In fact, the sections preceding and following this paragraph conceptually flow together easier.

3)      Is the discrepancy between the mRNA and protein expression levels of emerin due to increased emerin degradation, or increased post-transcriptional regulation of emerin mRNA? I feel that this point needs to be addressed in the text somewhere.

4)      All figures are high quality with the exception of Figure 1. Please check the conversion of this image as it appears to be at a much lower resolution than comparable figures in the manuscript, Eg. Compare Fig 1a with 3a.

5)      Is the decreased SUN2 expression a result of targeted protein degradation or inhibition of gene expression?

6)      Related to the previous question – is an intact LINC complex necessary for stress fiber formation? This should be included in the discussion, given the role of emerin at the nuclear envelope and with the nuclear lamins. Also, it may be useful to include the following citations:

a.       Versaevel, M. et al, Scientific Reports (2010) - Super-resolution microscopy reveals LINC complex recruitment at nuclear indentation sites.

b.      Maninova, M and Vomastek, T., FEBS Letters (2016) – Dorsal stress fibers, transverse actin arcs, and perinuclear actin fibers form an interconnected network that induces nuclear movement in polarizing fibroblasts.

7)      Is there a change in matrix deposition/turnover between cells with wild type vs mutant emerin?

Specific comments:

1)      Figure 1 – in addition to the low resolution issue mentioned earlier, I had a few technical questions and formatting comments.

a.       In Figure 1c, what happens for that 20 min period between the final and second last data points?

b.      In the electron microscopy images, to my untrained eye, the curvilinear structures described are much easier to see in one panel vs the other. Perhaps a schematic, or some arrows, to indicate to the reader what to look for may be useful

2)      Figure 6 – in Figure 6D, is this a western blot against phosphorylated emerin, or total emerin? Which bands are readers to focus on specifically? It appears that there may be some hyperphosophorylation occurring? If that is the case, what is the significance of the increased phosphorylation? Are there other kinases that phosphorylate emerin? Some clarification is required.

Author Response

The authors greatly thank Reviewer 2 for careful reading and constructive remarks. All comments have been taken into account in the reviised version. Here are the details concerning the changes we have made in this mansucript

General comments:

1)      There are a few syntax errors that interrupt the flow of the manuscript. For example, in the legend for Figure 1, it is written, “The three emerin mutations associated to cardiac defects…” should probably be revised to read, “The three emerin mutations associated with cardiac defects…”

Manuscript has been checked carefully. Syntax errors have been corrected.

2)      At least one paragraph in the introduction section reads more like it would fit in the Discussion section of the paper, i.e. – pg. 2, lines 49-64. In fact, the sections preceding and following this paragraph conceptually flow together easier.

We agree that we could displace this paragraph to the discussion. However, as our variants cause both a significant decrease in emerin protein level and a potential defect in partner binding due to the mutation, we think that it is important to introduce the general consequences of a reduction of emerin protein level before analyzing what is published about the impact of the specific mutations found in the variants. Most importantly, in the results section, we are referring to the emerin function in the cell response to a mechanical stress and this essential function being defective after RNAi-mediated depletion of emerin, it is introduced in this paragraph.

3)      Is the discrepancy between the mRNA and protein expression levels of emerin due to increased emerin degradation, or increased post-transcriptional regulation of emerin mRNA? I feel that this point needs to be addressed in the text somewhere.

We have added this point in the discussion (page 13) “Additionally, we observed a reduced emerin DK37 protein expression without any modification of emerin mRNA levels, indicating that emerin DK37 is either less transcribed because of defective post-transcriptional regulation of emerin mRNA, or degraded faster than wild-type emerin in myofibroblasts. Â»

4)      All figures are high quality with the exception of Figure 1. Please check the conversion of this image as it appears to be at a much lower resolution than comparable figures in the manuscript, Eg. Compare Fig 1a with 3a.

We modified Fig 1a in order to obtain a resolution comparable to that of the other panels and figures. We have added this point in the discussion

5)      Is the decreased SUN2 expression a result of targeted protein degradation or inhibition of gene expression?

This is an interesting point that deserves additional studies. Whether reduced SUN2 expression is a result of targeted protein degradation or inhibition of gene expression cannot be assessed from our study. For this, detailed protein regulation of SUN2 remains to be determined. The following sentence has been added in the Discussion (page 14): “Decreased SUN2 expression could result from targeted protein degradation or inhibition of gene expression and further studies are required to determine detailed protein regulation of SUN2."

6)      Related to the previous question – is an intact LINC complex necessary for stress fiber formation? This should be included in the discussion, given the role of emerin at the nuclear envelope and with the nuclear lamins. Also, it may be useful to include the following citations:

a.       Versaevel, M. et al, Scientific Reports (2010) - Super-resolution microscopy reveals LINC complex recruitment at nuclear indentation sites.

b.      Maninova, M and Vomastek, T., FEBS Letters (2016) – Dorsal stress fibers, transverse actin arcs, and perinuclear actin fibers form an interconnected network that induces nuclear movement in polarizing fibroblasts.

These comments have been taken into account. The following sentence has been added to the discussion (page 14): “The LINC complexes connect the perinuclear actin fibers to the nuclear envelope (Maninova et al., 2016). Tension in actin stress fibers leads to a local enrichment of LINC complexes and lamin A/C (Versaevel et al., 2014).”

7)      Is there a change in matrix deposition/turnover between cells with wild type vs mutant emerin?

This is an interesting point that cannot be precisely determined from our study given that cells were cultured in vitro. There was no real matrix deposition/turnover in vitro. Further studies are required to determine this point.

Specific comments:

1)      Figure 1 – in addition to the low resolution issue mentioned earlier, I had a few technical questions and formatting comments.

a.       In Figure 1c, what happens for that 20 min period between the final and second last data points?

There are 20 hours between these two data points. In fact, we captured the beginning of the kinetics at day 0, and recorded the last point on the next day.

b.      In the electron microscopy images, to my untrained eye, the curvilinear structures described are much easier to see in one panel vs the other. Perhaps a schematic, or some arrows, to indicate to the reader what to look for may be useful

We have changed the EM images to display images obtained with more similar staining and background, and we have added an EM image of the wild-type filaments for comparison.

2)      Figure 6 – in Figure 6D, is this a western blot against phosphorylated emerin, or total emerin? Which bands are readers to focus on specifically? It appears that there may be some hyperphosphorylation occurring? If that is the case, what is the significance of the increased phosphorylation? Are there other kinases that phosphorylate emerin? Some clarification is required.

Figure 6D is an SDS-PAGE gel, so the proteins are revealed using Coomassie Blue. The gel is a bit overloaded so that the very intense bands correspond to EmN (either WT or mutated) and the weaker bands correspond to contaminants present in very low amounts as compared to EmN (these contaminants are bacterial proteins given that EmN is expressed in and purified from bacteria). We have now added two arrows indicating where EmN migrates before and after phosphorylation. EmN is incubated in the presence of Src, so in this experiment it is only phosphorylated by Src. However, we have shown by NMR that EmN can also be phosphorylated by a large set of kinases, as CK2, PLK1 and P38. These experiments are out of the scope of this study.